# High-Precision Atom Interferometer-Based Dynamic Gravimeter Measurement by Eliminating the Cross-Coupling Effect

**DOI:** 10.3390/s24031016

**Published:** 2024-02-04

**Authors:** Yang Zhou, Wenzhang Wang, Guiguo Ge, Jinting Li, Danfang Zhang, Meng He, Biao Tang, Jiaqi Zhong, Lin Zhou, Runbing Li, Ning Mao, Hao Che, Leiyuan Qian, Yang Li, Fangjun Qin, Jie Fang, Xi Chen, Jin Wang, Mingsheng Zhan

**Affiliations:** 1Innovation Academy for Precision Measurement Science and Technology, Chinese Academy of Sciences, Wuhan 430071, China; zhouyang@apm.ac.cn (Y.Z.); wangwz@apm.ac.cn (W.W.); geguiguo@apm.ac.cn (G.G.); lijinting@apm.ac.cn (J.L.); zhangdf@apm.ac.cn (D.Z.); hemeng@apm.ac.cn (M.H.); biaotang@apm.ac.cn (B.T.); jqzhong@apm.ac.cn (J.Z.); lzhou@apm.ac.cn (L.Z.); rbli@apm.ac.cn (R.L.); wangjin@apm.ac.cn (J.W.); mszhan@apm.ac.cn (M.Z.); 2University of Chinese Academy of Sciences, Beijing 100049, China; 3Wuhan Institute of Quantum Technology, Wuhan 430206, China; 4Hefei National Laboratory, Hefei 230094, China; 5School of Electrical Engineering, Naval University of Engineering, Wuhan 430033, China; maoningbit@163.com (N.M.); hg15441@163.com (H.C.); hjgcdxqly@126.com (L.Q.); hjgcdxly@126.com (Y.L.)

**Keywords:** atom interferometer, dynamic gravimeter measurement, high precision, gravimeter, marine gravity survey, cold atom

## Abstract

A dynamic gravimeter with an atomic interferometer (AI) can perform absolute gravity measurements with high precision. AI-based dynamic gravity measurement is a type of joint measurement that uses an AI sensor and a classical accelerometer. The coupling of the two sensors may degrade the measurement precision. In this study, we analyzed the cross-coupling effect and introduced a recovery vector to suppress this effect. We improved the phase noise of the interference fringe by a factor of 1.9 by performing marine gravity measurements using an AI-based gravimeter and optimizing the recovery vector. Marine gravity measurements were performed, and high gravity measurement precision was achieved. The external and inner coincidence accuracies of the gravity measurement were ±0.42 mGal and ±0.46 mGal after optimizing the cross-coupling effect, which was improved by factors of 4.18 and 4.21 compared to the cases without optimization.

## 1. Introduction

Gravity measurements have important applications in various fields, such as geodesy, geophysics [1], navigation [2], and fundamental physics tests [3,4]. Gravity can be obtained from static, dynamic, and satellite measurements. Dynamic gravimeters rely on dynamic carriers, such as ships and aircraft. They can obtain gravitational information accurately and efficiently and are usually relative sensors. They suffer from drift and have to be calibrated before and after the gravity measurement. Recently, dynamic gravimetry based on atomic interferometry has been developed [5,6]. It can measure absolute gravity without measurement drift, which has wide potential applications.

The time-pulsed atom interferometer (AI) was first realized in 1991 [7] and has been widely used for precision gravity measurements [8,9,10,11,12,13,14,15], gravity gradient measurements [16], and rotation measurements [17]. Dynamic gravity measurements based on AI have been realized in a moving elevator [18], vehicles [19,20,21,22], aircraft [23,24,25] and ships [6,26,27,28], and the best measurement precision is better than 1 mGal [6,27]. Various methods, such as vibration compensation [29,30] and data filtering [31,32,33], have been proposed to further improve the precision of AI-based dynamic gravimeters.

An AI-based dynamic gravimeter is a type of joint measurement using an AI and a classical accelerometer. The AI utilizes the Raman laser to measure the acceleration of the free-falling cold atom cloud relative to the retro-reflecting mirror of the Raman laser. It has the advantage of absolute and high-precision gravity measurement, but suffers from low data sample rate and limited measurement range. The classical accelerometer, which is mounted to the Raman laser’s mirror, measures the acceleration of the mirror itself. It has the advantages of a wide measurement range and high sample rate for gravity measurement but has the disadvantage of unpredictable measurement drift. In the method of joint measurement, the output of a classical accelerometer is compared and corrected by the measured acceleration of the AI in real time. The bias and drift of a classical accelerometer are eliminated. This method benefits from both the advantages of the AI and the classical accelerometer and can provide accurate and continuous gravity outputs with a wide measurement range. The dynamic environment can degrade the precision of gravity measurement. Additional noise will be induced if the sensitive axes of the classical accelerometer are misaligned with the sensitive axes of the AI. We call this effect the cross-coupling effect. This effect could be induced by the installation error between the classical accelerometer and the AI, and the cross talk of different sensitive axes of the classical accelerometer itself [30].

In this study, we analyzed the cross-coupling effect and developed a method to eliminate it. Subsequently, we performed marine gravity measurements and achieved a high measurement precision. The remainder of this paper is organized as follows: In Section 2, we introduce the cross-coupling effect and analyze the induced phase noise. A recovery vector is proposed to eliminate this phase noise. In Section 3, we introduce the marine gravity measurements experiment by using an AI-based gravimeter. Gravity measurement results with and without optimizing the cross-coupling effect are evaluated and compared. In Section 4, conclusions and discussion are presented.

## 2. Theoretical Methods

### 2.1. Joint Gravity Measurement Process

The principle of the joint gravity measurement process is illustrated in Figure 1. The acceleration felt by AI is $\overset{⃑}{a}\left(t\right)=\{a_{x}\left(t\right),a_{y}\left(t\right),a_{z}\left(t\right)\}$, and $a_{z}\left(t\right)$ is in the direction of gravity. The AI measures $a_{z}\left(t\right)$ with its sensitive axis adjusted to the z-axis. The classical accelerometer can measure the acceleration in 3 axes, and the measured acceleration is ${\overset{⃑}{a}}_{\mathrm{cla}}\left(t\right)=\{a_{\mathrm{cla},x}\left(t\right),a_{\mathrm{cla},y}\left(t\right),a_{\mathrm{cla},z}\left(t\right)\}$. In the ideal situation, the sensitive axes of the accelerometer coincide with the axes of $\overset{⃑}{a}\left(t\right)$, and the value of ${\overset{⃑}{a}}_{\mathrm{cla}}\left(t\right)$ is different from $\overset{⃑}{a}\left(t\right)$ with an offset acceleration ${\overset{⃑}{a}}_{\mathrm{off}}\left(t\right)=\{a_{\mathrm{off},x}\left(t\right),a_{\mathrm{off},y}\left(t\right),a_{\mathrm{off},z}\left(t\right)\}$, which is caused by the preset offset or the measurement drift of the accelerometer. Then, we have
(1)$${\overset{⃑}{a}}_{\mathrm{cla}}\left(t\right)=\overset{⃑}{a}\left(t\right)+{\overset{⃑}{a}}_{\mathrm{off}}\left(t\right).$$

The population of AI after an interference measurement circle is *P*. It has the following relationship with the interference phase $\phi_{\mathrm{AI}}$:(2)$$P=P_{0}\left[1+b\cos\left(\phi_{\mathrm{AI}}\right)\right],$$
where $P_{0}$ and *b* are the offset and contrast of the interference fringe. The interference phase can be calculated as
(3)$$\phi_{\mathrm{AI}}=k_{\mathrm{eff}}\int_{-T}^{T}f\left(t\right)[a_{z}\left(t\right)-a_{\mathrm{chirp}}]\mathrm{dt},$$
where $k_{\mathrm{eff}}$ is the effective wave vector, 2*T* is the interference time, f(t) is the response function [23], $a_{ch\mathrm{irp}}=2\pi \alpha_{0}/k_{\mathrm{eff}}\;$ is a constant equivalent acceleration induced by the laser frequency chirp, and $\alpha_{0}$ is the laser frequency chirp rate used to compensate for the Doppler shift of the atom cloud. Due to the vibration noise component in $\overset{⃑}{a}\left(t\right)$, the interference phase $\phi_{\mathrm{AI}}$ and the population *P* will vary randomly. The so-called vibration compensation method is employed to recover the interference fringe. The acceleration component $a_{\mathrm{cla},z}\left(t\right)$ is used to calculate the compensation phase $\phi_{\mathrm{com}}$ as
(4)$$\phi_{\mathrm{com}}=k_{\mathrm{eff}}\int_{-T}^{T}f\left(t\right)a_{\mathrm{cla},z}\left(t\right)\mathrm{dt}.$$

Then, $\phi_{\mathrm{com}}$ and *P* are set as the *x*- and *y*-axis coordinates, respectively, to recover the interference fringe. By substituting Equations (1), (3), and (4) into Equation (2), one has
(5)$$P=P_{0}\left[1+b\cos\left(\phi_{\mathrm{com}}-\phi_{\mathrm{fit}}\right)\right],$$
where we call $\phi_{\mathrm{fit}}=\phi_{\mathrm{com}}-\phi_{\mathrm{AI}}$ the fitting phase. By fitting this fringe, one can obtain the fitting phase $\phi_{\mathrm{fit}}$, and thus the so-called fitting acceleration $a_{\mathrm{fit}}\equiv \phi_{\mathrm{fit}}/k_{\mathrm{eff}}T^{2}$. The fitting phase has the form
(6)$$\phi_{\mathrm{fit}}={\;k}_{\mathrm{eff}}\int_{-T}^{T}\;f\left(t\right)[a_{\mathrm{off},z}\left(t\right)+a_{chirp}]\mathrm{dt}.$$

From Equation (6), we can see that the fitting phase is not coupled with the acceleration, which means that the vibration compensation method is perfect and the acceleration will not induce noise to the fitting phase. If ${\overset{⃑}{a}}_{\mathrm{off}}\left(t\right)$ is constant, one can derive that $a_{\mathrm{fit}}=a_{\mathrm{off},z}+a_{chirp}$. By substituting this relationship into Equation (1), one can obtain
(7)$$a_{z}\left(t\right)=a_{\mathrm{cla},z}\left(t\right)\;-a_{\mathrm{fit}}+a_{\mathrm{chirp}}.$$

The acceleration $a_{z}\left(t\right)$ felt by AI contains the acceleration of gravity $a_{\mathrm{gra}}\left(t\right)$ and the motion acceleration $a_{\mathrm{mot},z}\left(t\right)$ of the gravimeter at the z direction. The gravitational acceleration $a_{\mathrm{gra}}\left(t\right)$ can be calculated as
(8)$$a_{\mathrm{gra}}\left(t\right)=a_{\mathrm{cla},z}\left(t\right)-\;a_{\mathrm{fit}}+a_{\mathrm{chirp}}-a_{\mathrm{mot},z}\left(t\right).$$

The acceleration $a_{\mathrm{cla},z}\left(t\right)$ is measured by the classical accelerometer. The motion acceleration $a_{\mathrm{mot},z}\left(t\right)$ can be calculated from the position information of the gravimeter, for example, from the signal of the Global Navigation Satellite System (GNSS) [24]. The noises of these two terms can be reduced by using the low-pass filter. $a_{\mathrm{chirp}}$ is a constant acceleration. This article focuses on the noise of fitting acceleration $a_{\mathrm{fit}}$ caused by the cross-coupling effect.

### 2.2. Noise Induced by the Cross-Coupling Effect and Introduction of the Recovery Vector

We introduce the cross-coupling effect for the case that the sensitive axes of the accelerometer are different from $\overset{⃑}{a}\left(t\right)$ and describe this effect as the following relationship
(9)$${\overset{⃑}{a}}_{\mathrm{cla}}\left(t\right)=C\cdot \overset{⃑}{a}\left(t\right)+{\overset{⃑}{a}}_{\mathrm{off}}\left(t\right),$$
where ***C*** $=\left[C_{i,j}\right]\left(i,j=x,y,z\right)$ is the coupling matrix. If we still use the z-component of the classical accelerometer $a_{\mathrm{cla},z}\left(t\right)\;$ to calculate the compensation phase, after some derivation, the fitting phase has the form
(10)$$\phi_{\mathrm{fit}}=k_{\mathrm{eff}}\int_{-T}^{T}f\left(t\right)[\overset{⃑}{c}\cdot \overset{⃑}{a}\left(t\right)+a_{\mathrm{off},z}\left(t\right)-a_{z}\left(t\right)+a_{chirp}]\mathrm{dt},$$
where we define $\overset{⃑}{c}\equiv \left\{C_{z,x},C_{z,y},C_{z,z}\right\}$ as the coupling vector. One can see from Equation (10) that if $\overset{⃑}{c}\neq \{0,0,1\}$, the acceleration noise will couple to the coupling vector and induce phase noise for the fitting phase.

To reduce this phase noise, we propose that a recovering process be inserted before the calculation of the compensation phase, as shown in Figure 1. A matrix ***D*** = $\left[D_{i,j}\right]\;(i,j=x,y,z)$ is introduced to recover the measured acceleration of the classical accelerometer as
(11)$${\overset{⃑}{a}}_{\mathrm{rec}}\left(t\right)=D\cdot {\overset{⃑}{a}}_{\mathrm{cla}}\left(t\right).$$

We call ${\overset{⃑}{a}}_{\mathrm{rec}}\left(t\right)$ the recovery acceleration, and we still use the z component of ${\overset{⃑}{a}}_{\mathrm{rec}}\left(t\right)$ to calculate the compensation phase; after some derivation, the fitting phase has the following form
(12)$$\phi_{\mathrm{fit}}=k_{\mathrm{eff}}\int_{-T}^{T}f(t)[\overset{⃑}{d}\cdot (C\cdot \overset{⃑}{a}\left(t\right))+{\overset{⃑}{d}\cdot \overset{⃑}{a}}_{\mathrm{off}}\left(t\right)-a_{z}\left(t\right)+a_{\mathrm{chirp}}]\mathrm{dt},$$
where we define the recovery vector as $\overset{⃑}{d}\equiv \{D_{z,x},D_{z,y},D_{z,z}\}$ and $a_{\mathrm{rec},z}\left(t\right)=\overset{⃑}{d}\cdot {\overset{⃑}{a}}_{\mathrm{cla}}\left(t\right)$. For the special case of $D=C^{-1}$, the fitting phase has the form
(13)$$\phi_{\mathrm{fit}}=k_{\mathrm{eff}}\int_{-T}^{T}f(t)[\overset{⃑}{d}\cdot {\overset{⃑}{a}}_{\mathrm{off}}\left(t\right)+a_{chirp}]\mathrm{dt}.$$

One can see that, similar to Equation (6), the acceleration will not induce noise to the fitting phase. However, for the general form of the recovery vector, the acceleration noise will induce phase noise to the fitting phase. The problem is how to optimize the recovery vector $\overset{⃑}{d}$ for unknown coupling matrix C. The detailed optimization process will be described in Section 3.4. We also want to mention that, because of the introduction of the recovering process, the formulas presented in Equations (7) and (8) to calculate the acceleration and gravity should make some modifications. They are modified as
(14)$$a_{z}\left(t\right)=a_{\mathrm{rec},z}\left(t\right)\;-a_{\mathrm{fit}}+a_{\mathrm{chirp}}.$$
(15)$$a_{\mathrm{gra}}\left(t\right)=a_{\mathrm{rec},z}\left(t\right)-a_{\mathrm{fit}}+a_{\mathrm{chirp}}-a_{\mathrm{mot},z}\left(t\right).$$

## 3. Marine Gravity Measurement Experiment

### 3.1. Experiment Apparatus

We developed a compact AI-based dynamic gravimeter, as shown in Figure 2. It consists of the sensor head [28], the inertial stabilization platform [33], the optical system [15,34], and the electronic system [15]. Atomic interference occurs at the sensor head. It utilizes the rubidium-85 cold atom cloud as the test mass, and it is surrounded by a magnetic field shield. Additionally, a classical accelerometer (Titan accelerometer from Nanometrics [35]) is mounted on top of it. The sensor head has a compact size of 17 cm × 17 cm × 60 cm and a weight of only 15 kg. It is installed on a homemade dual-axis inertial stabilization platform. The platform has a size of ϕ58 cm × 100 cm and a weight of about 150 kg and offers an angle control accuracy of approximately 0.2 mrad with a load of 30 kg [33]. A compact optical system is used to provide the required laser power. It consists of several homemade fiber modules and occupies a 3U chassis. Two laser beams are sent to the sensor head by two single-mode polarization-maintaining fibers [15,34]. The electronic system is used to drive the components of the gravimeter, generate the time sequence, and acquire and process the experimental data [15]. A GNSS receiver is used to obtain the position information.

The process of the AI is similar to the process of the AI-based gravimeter in [15,34]. Here, we introduce the process briefly. Firstly, the cold atom cloud is obtained by a three-dimensional magneto-optical trap (3D-MOT) and polarization gradient cooling (PGC). The number of the cold atoms is about 10^7^ and the temperature of the cold atoms is about 5 μK. Then, the cold atoms are prepared to the |5^2^S_1/2_, F = 2> state by applying the state preparation laser pulse. Secondly, the π/2-π-π/2 Raman pulses are applied to realize the Raman interference process. The time interval between Raman pulses can be adjusted from 10 ms to 40 ms. Finally, the population of the cold atoms is obtained by the normalized fluorescence detection. The free fall distance of the cold atom is about 184 mm and the circle time for each measurement is about 600 ms.

### 3.2. Systematic Error Evaluation of the AI-Based Gravimeter

Before the marine gravity measurements, we performed a systematic error evaluation of the AI-based gravimeter. Long-term gravity measurements were conducted at the National Geodetic Observatory in Wuhan. The interference time was set to 2T = 30 ms, which was the same as that in the dynamic case. Several systematic error terms were evaluated, as listed in Table 1. The gravity gradient term was evaluated using the local gravity gradient and height of the sensor head. The single- and double-photon light shift terms were evaluated using the sideband ratio of the Raman laser and the time sequence of the Raman laser pulses [36]. The multi-sideband feature of the Raman laser induced an additional laser line effect because a fiber electro-optic modulator (FEOM) was used to produce the Raman laser. This effect was evaluated using the sideband ratio of the Raman laser, the position of the reflection mirror of the Raman laser, and the trajectory of the cold atom cloud [37]. The solid tide term was evaluated using theoretical calculations. The wave vector inversion method was adopted to suppress the systematic errors induced by the Zeeman and AC Stark shifts. After the systematic error correction and the long time gravity measurement, the measured gravity of the AI-based gravimeter, compared with the gravity value of the reference site, still had an offset of 116 μGal. This offset might be caused by the residual Zeeman shift, wavefront aberration of the Raman laser, or other systematic error terms. We treated this offset as a calibration term, as listed in Table 1. We deduced the solid-tide-induced gravity variation from the measured gravity and calculated the Allan standard deviation, as shown in Figure 3. The gravity measurement resolution was approximately 1.85 mGal and 0.05 mGal at 1 s and 5000 s, respectively. More details can be found in the reference [8]. For the dynamic case, due to the mean value of the external acceleration being zero, the averaged systematic error in Table 1 will not change.

### 3.3. Gravity Comparison under the Mooring State

The gravimeter was then transferred from Wuhan to Zhejiang Province and installed on a survey ship. Before and after the dynamic gravity survey, we compared the gravity measurement values of an AI-based gravimeter with the gravity value of a shore-based gravity reference site when the survey ship was moored to the dock. We call this state the mooring state. The latitude and height differences between the gravimeter and the reference site were measured, and the induced gravity difference was calculated and compensated for the measured gravity ([App appA-sensors-24-01016]). During each comparison, we measured gravity for 40 min and compared the average gravity value with that of the reference site. The measured gravity differences are shown in Figure 4. The data points had a mean value of −0.32 mGal and a standard deviation of 0.22 mGal. No apparent drift was observed before or after the dynamic gravity surveying. The measurement errors before and after the gravity survey are different; this is mainly caused by the different sea conditions.

### 3.4. Gravity Measurement under the Sailing State

Marine gravity measurements were conducted in the East China Sea. We carried out repeated survey measurements in the east–west direction. The number of survey lines was three. We removed some data points around the turning points of the survey lines due to the unstable survey speed. The effective length of a single survey line was 45 km, and the survey speed was approximately 11 knots. The trajectory of the survey line is shown in Figure 5a. The power spectral density (PSD) of the acceleration was measured by the Titan accelerometer during the survey and in the mooring state, as shown in Figure 5b. The accelerations measured along the survey line had a peak-to-peak value of approximately 0.6 m/s^2^. In order to keep the measurement precision while adapting to the dynamic sea conditions, the interference time was set to 2T = 30 ms during the gravity measurement. A classical shipborne strapdown gravimeter (SN-022) [33,38] was installed nearby for gravity measurements comparison.

Before processing the measured gravity data, we set a value for the recovery vector and recovered the interference fringe, as described in Section 2.2. The recovered fringe was fitted to obtain the fitting phase $\phi_{\mathrm{fit}}$, and a group of the fitting phases were obtained to calculate their standard deviation $\sigma_{\phi \text{\_}\mathrm{fit}}$. We scanned the values of the components of the recovery vector $\overset{⃑}{d}\;$ around {0, 0, 1} and found the relationship between $\sigma_{\phi \text{\_}\mathrm{fit}}$ and the recovery vector components. The corresponding curves are referred to be the calibration curves, as shown in Figure 6a. These curves had a valley shape, and the widths of the valleys were inversely proportional to their corresponding coupling accelerations. This can be understood using Equation (12). The recovery vector is coupled to the acceleration. If the coupled acceleration is small, an offset of the recovery vector from its optimized value will lead to small phase noise. The x-coordinates of the bottom of the valleys represent the optimized values of the components of the recovery vector. The optimized recovery vector during the survey measurement was found to be {0.0060, −0.0034, 0.9860}. The typical recovered fringes before and after the optimization are shown in Figure 6b,c. The fitting phase uncertainty of these two fringes was 0.19 rad and 0.10 rad, respectively. The phase noise was improved by a factor of 1.9.

After the optimization of the recovery vector, we calculated the gravity along the survey lines. Firstly, the optimized recovery vector was used to calculate the recovery acceleration $a_{\mathrm{rec},z}\left(t\right)$. The measured value of $a_{\mathrm{rec},z}\left(t\right)$ during the gravity survey is shown in Figure 7a. Secondly, $a_{\mathrm{rec},z}\left(t\right)$ was filtered using a fourth-order Bessel low-pass filter to filter the motion acceleration of the surveying ship. The time constant of the filter was set to be 300 s, and the filtered acceleration is shown in Figure 7b. Thirdly, the motion acceleration, $a_{mot,z}\left(t\right)$, was calculated using the recorded GNSS signal. The same low-pass filter was used for this acceleration. The filtered motion acceleration is shown in Figure 7c. Then, the vibration compensation method was applied to recover the interference fringe. By fitting the phase of the fringe, the acceleration $a_{\mathrm{fit}}$ was obtained. Finally, Equation (15) was used to calculate the acceleration of gravity $a_{\mathrm{gra}}\left(t\right)$.

The time-varied solid tide-induced gravity was calculated and subtracted from $a_{\mathrm{gra}}\left(t\right)$. In order to subtract the majority part of the gravity and show the gravity change, the normal gravity (model is China Geodetic Coordinate System 2000) was subtracted from $a_{\mathrm{gra}}\left(t\right)$ to obtain the gravity anomaly. The calculated gravity anomaly along the survey lines is shown in Figure 7d. This gravity anomaly was compared with the gravity anomaly measured by the SN-022 shipborne strapdown gravimeter. The external coincidence accuracies of the gravity anomaly measurements ([App appB-sensors-24-01016]) for the three survey lines were calculated. The results were ±0.46 mGal, ±0.42 mGal, and ±0.41 mGal, respectively. The result for the three lines in total was ±0.43 mGal. However, if the recovery vector was set to be {0, 0, 1}, the calculated external coincidence accuracy of the three lines was found to be ±1.80 mGal. The gravity measurement accuracy was improved by a factor of 4.18 by optimizing the cross-coupling effect.

Then, we calculated the inner coincidence accuracy of the gravity measurements of the three survey lines ([App appB-sensors-24-01016]). The time-varying gravity anomaly data were converted to position-varying data along the survey lines. The results are shown in Figure 8a. Significant gravity measurement deviations were observed across the three survey lines. This was not mainly caused by the measurement offset of the AI-based gravimeter but by the fluctuations in the sea surface height. To eliminate this effect, the water depth was measured in real time, as shown in the inset figure of Figure 8b, and the height-induced gravity variation was calculated. This gravity variation was deduced from the measured gravity anomaly. The measured gravity anomaly was transferred from the surface to the bottom of the sea. Then, the gravity anomalies of the three survey lines were compared again, and the results are shown in Figure 8b. The consistency of the gravity anomalies was better than that in Figure 8a, and the inner coincidence accuracy is calculated to be ±0.46 mGal. If the recovery vector was set to be {0, 0, 1}, the calculated inner coincidence accuracy was found to be 1.9 mGal. The gravity measurement accuracy was improved by a factor of 4.21.

## 4. Conclusions

In this study, we introduced a model of a joint gravity measurement process for AI-based dynamic gravity measurements. The cross-coupling effect was analyzed, and a recovery vector was introduced to suppress this effect. The phase noise of the interference fringe was improved by a factor of 1.9 in the sailing state by optimizing the value of the recovery vector. Subsequently, the design of an AI-based dynamic gravimeter was introduced, which was used for marine gravity measurement. Before the gravity survey, the systematic error of the AI-based gravimeter was evaluated at a gravity observatory. A gravity comparison with the shore-based gravity reference was performed in the mooring state. The measured gravity difference had a mean value of −0.32 mGal and a standard derivation of 0.22 mGal. Marine gravity measurements were performed using repeated survey lines. The measured gravity anomaly was compared with that of a classical shipborne strapdown gravimeter. After optimizing the recovery vector, we achieved high precision for the dynamic gravity measurement. The external coincidence accuracies of the three survey lines in total was ±0.43 mGal. The gravity measurement inner coincidence accuracy of the three survey lines was ±0.46 mGal after considering the water depth-induced gravity variation. By optimizing the cross-coupling effect, the gravity measurement external and inner coincidence accuracies were improved by factors of 4.18 and 4.21.

The introduction and optimization of the recovery vector are important for high-precision AI-based marine gravity measurements. We believe that the strategies presented in this study will be beneficial for the future design and data analysis of AI-based dynamic gravimeters. Further improvements to the dynamic gravity measurement precision include the accurate calibration of the transfer function of the classical accelerometer and the analysis of the cold atom cloud trajectory under dynamic environments.

## Figures and Tables

**Figure 1 sensors-24-01016-f001:**
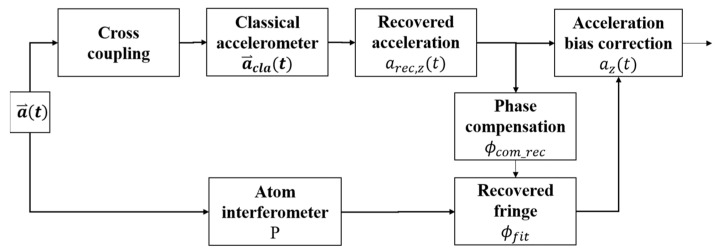
Principle of the joint gravity measurement and the introduction of the cross-coupling effect.

**Figure 2 sensors-24-01016-f002:**
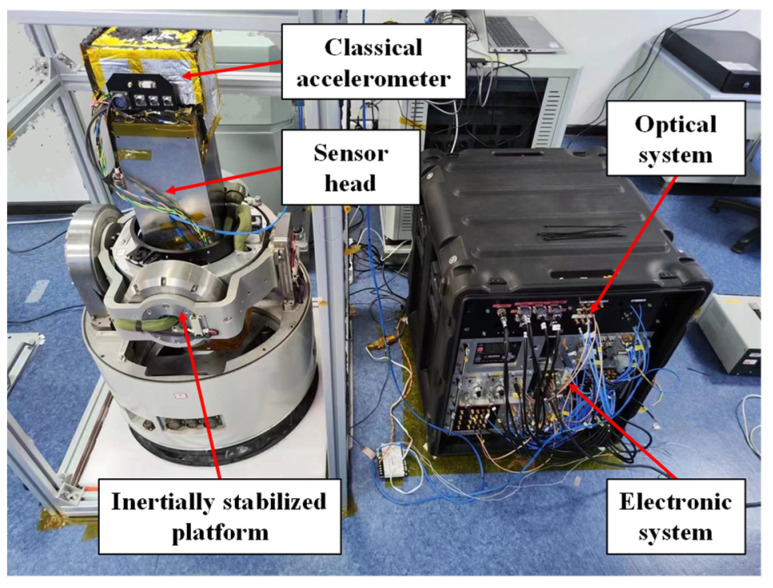
(Color online). AI-based dynamic gravimeter for the marine gravity measurement.

**Figure 3 sensors-24-01016-f003:**
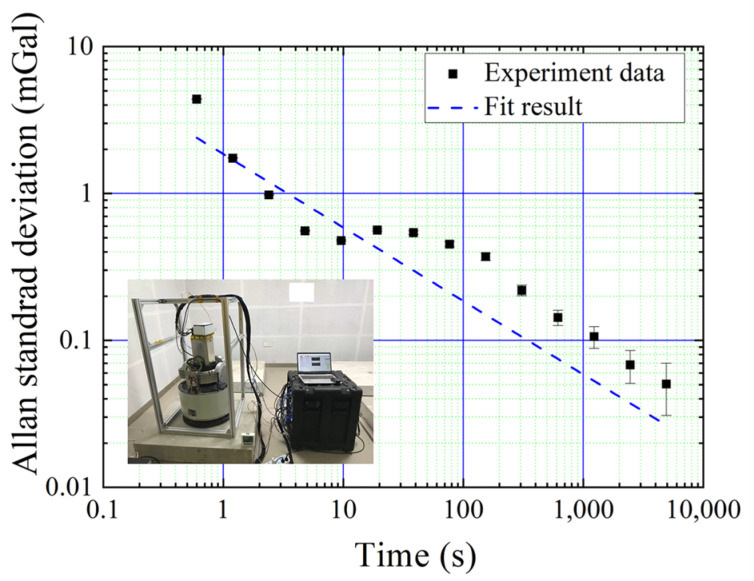
(Color online.) Allan standard deviation of the measured gravity value at the National Geodetic Observatory in Wuhan for 2T = 30 ms.

**Figure 4 sensors-24-01016-f004:**
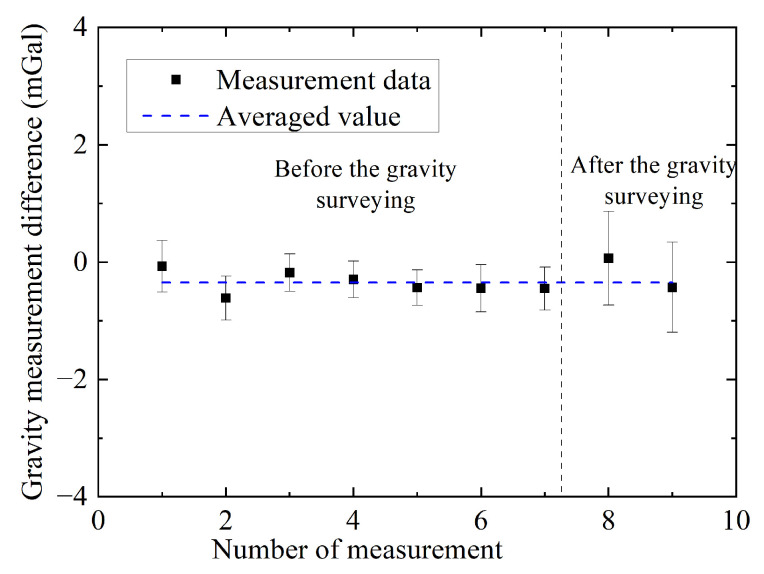
(Color online). Gravity comparison with a shore-based gravity reference site under mooring state.

**Figure 5 sensors-24-01016-f005:**
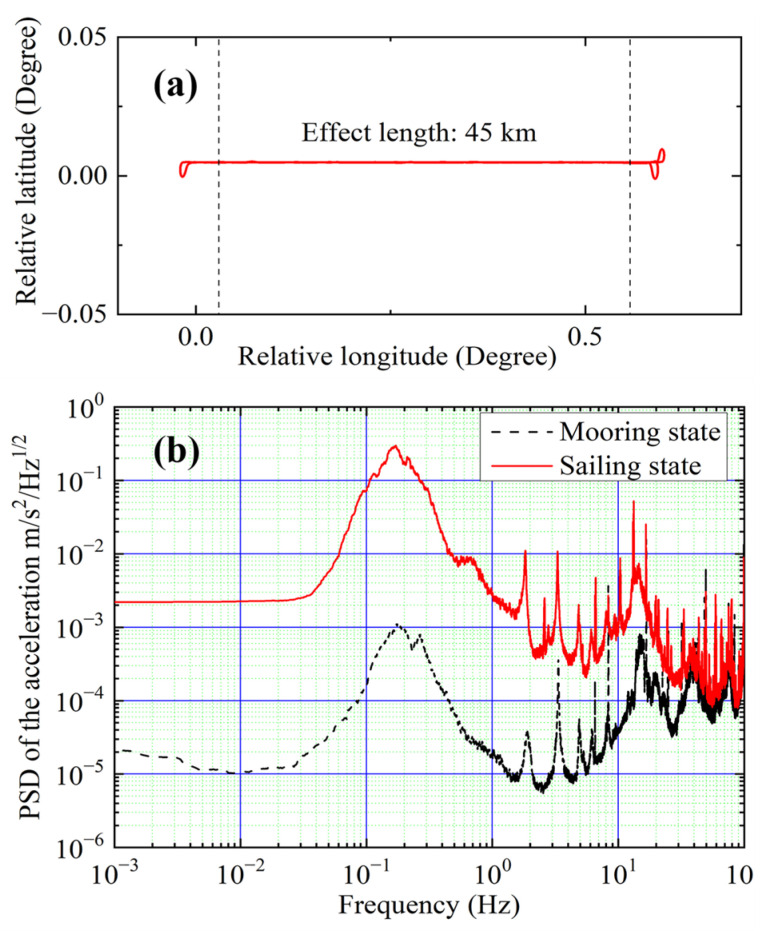
(Color online.) (**a**) The trajectory of the survey line during the marine gravity measurement. (**b**) The power spectral density amplitude of the measured acceleration in the z direction under the mooring state (black dashed line) and sailing state (red solid line).

**Figure 6 sensors-24-01016-f006:**
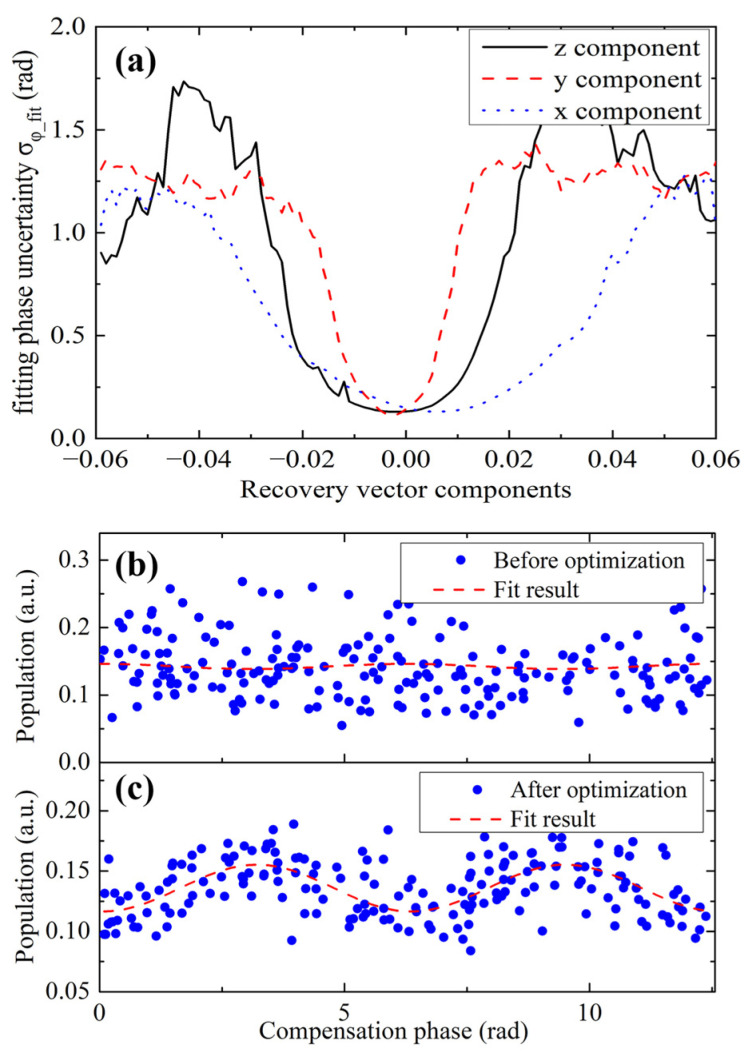
(Color online). (**a**) The calibration curves for the recovery vector components during the gravity survey measurement, the z component of recovery vector is subtracted by 0.99 for the convenience of display. (**b**) The recovered fringe when the recovery vector is set to {0, 0, 1}. (**c**) The recovered fringe when the recovery vector is set to its optimized value {0.0060, −0.0034, 0.9860}.

**Figure 7 sensors-24-01016-f007:**
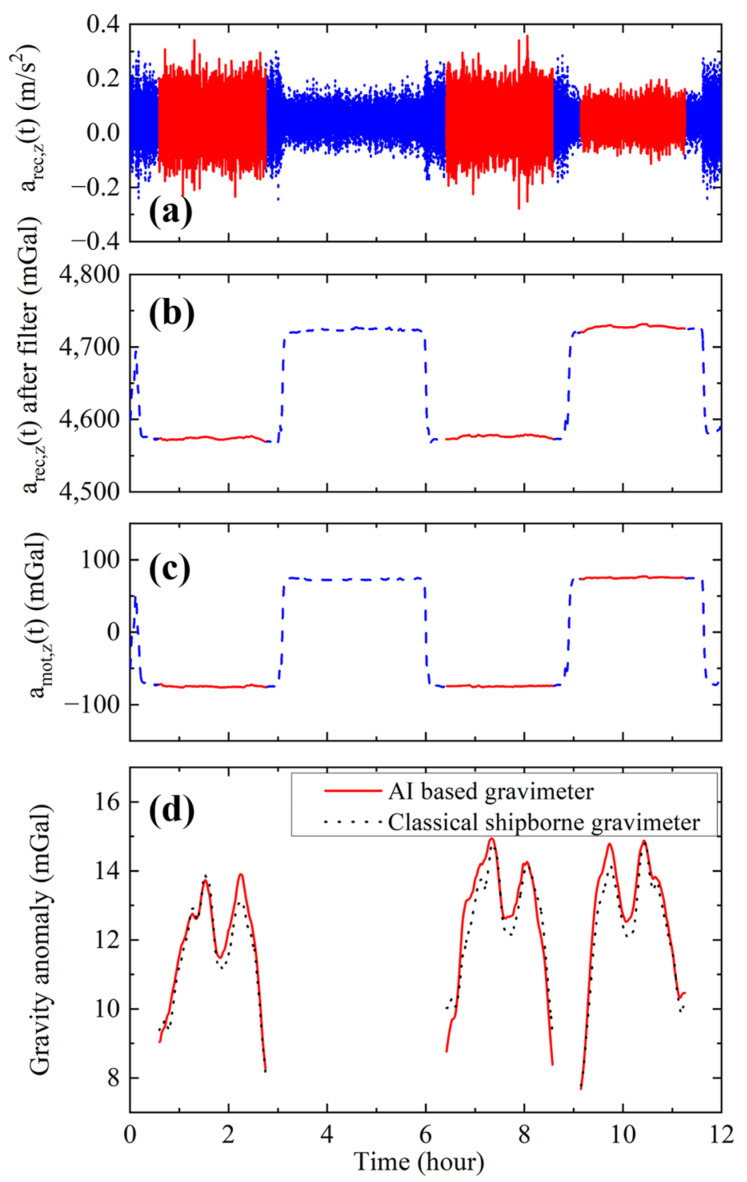
(Color online). Data processing process of the gravity anomaly. The red solid line in (**a**–**c**) represents data of the three survey lines, while the blue dashed line represents other data during the gravity survey. (**a**) The recovery acceleration *a_rec_*_,*z*_(*t*). (**b**) The recovery acceleration *a_rec_*_,*z*_(*t*) after the low-pass filter. (**c**) The calculated motion acceleration $a_{\mathrm{mot},z}(t)$ after the low-pass filter. (**d**) The measured gravity anomaly of the AI-based gravimeter (red solid line) and the classical shipborne strapdown gravimeter (black dotted line).

**Figure 8 sensors-24-01016-f008:**
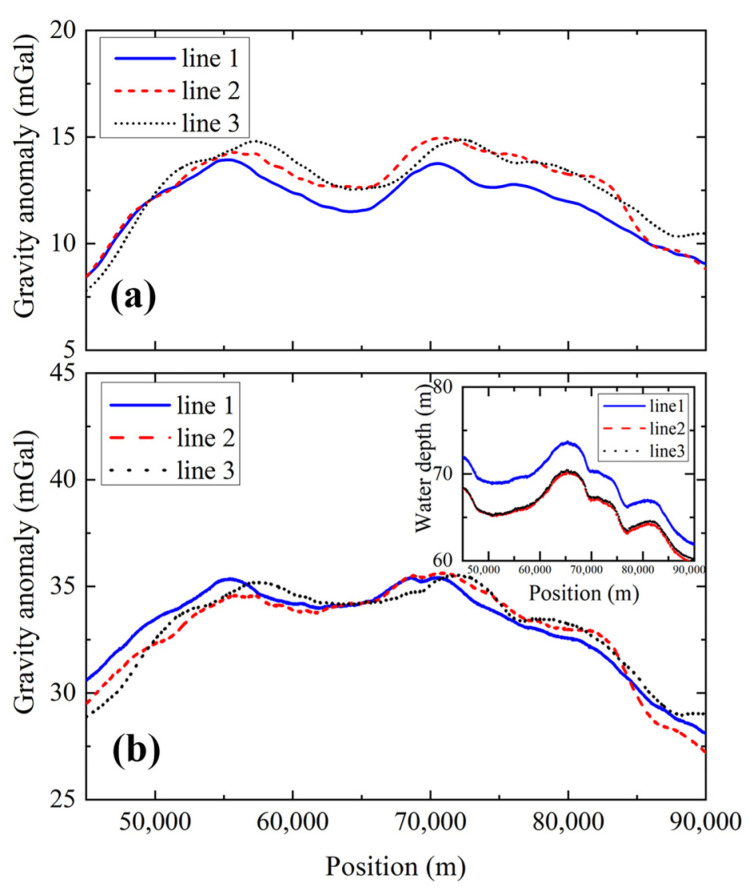
(Color online). Comparing the gravity anomaly measurement during the three survey lines. (**a**) Before deducting the sea surface height-induced gravity. (**b**) After deducting the sea surface height-induced gravity. The inset figure is the measured water depth during the three survey lines.

**Table 1 sensors-24-01016-t001:** Systematic error evaluation for the AI-based gravimeter.

Systematic Error Terms	Value (mGal)	Uncertainty (mGal)
Gravity gradient	−0.222	0.002
Single photon light shift	0.000	0.008
Double photon light shift	0.047	0.005
Additional laser lines	−0.699	0.137
Gravity calibration	−0.116	0.050
Systematic error in total	−0.990	0.147

## Data Availability

Data supporting the findings of this study are available from the corresponding author upon reasonable request. The data are not publicly available due to the importance of the high-precision gravity data.

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
