# Peer review of "High-Precision Atom Interferometer-Based Dynamic Gravimeter Measurement by Eliminating the Cross-Coupling Effect"

_sensors, 2024, doi:10.3390/s24031016_

Round 1

Reviewer 1 Report

Comments and Suggestions for Authors

The Editor may feel differently, but I suggest that a bit more background information be given so that the paper can be of interest to a broader readership than just experts in hybrid Raman gravitational interferometry.

For clarity, all abbreviations should be defined near their point of first use.

What is meant by "internal" and "external" coincidences? What is their intuitive physical meaning?

Explanations or citations should be given for all procedures that are used.

Some specific comments:

lines 4-6: the order of authors seems odd in that it is far from alphabetical and looks essentially random.

l. 22: clarify whether the claimed improvement factors of about 4.2 are already taken into account in the claimed accuracies. I believe they are, but the language used is ambiguous in this regard.

Introduction, paragraph 1: Please give a bit more background about the techniques employed. For example, in what way does an AI complement a classical accelerometer? What are the advantages and disadvantages of a hybrid approach using both? This can be brief - just a few more sentences or lines would make a big difference in making the paper approachable to students and interested readers from other specialties.

l. 44: non sequitur. Some smooth transition is needed, perhaps by adding another sentence or two between the first and second sentences. 1st sentence is purely introductory, leading reader to expect next a further description or discussion of the technique, not a statement of what was done, which is what the 2nd and 3rd sentences are. Also, it seems inappropriate and potentially confusing to mix tenses as done here: 1st sentence is present tense, 2nd and 3rd past, 4th present again.

l. 47: "However..." - non sequitur

Fig. 1: labels inconsistent - "Recover" vs. "Recovering" - but neither seems correct English

l. 65: "joined"->"joint"

l. 71: "series data"->"data series"

l. 75: "interferometers" - why plural?

l. 78: why does chirp induce apparent gravitational acceleration? Why is it constant?

l. 79: "gravity" - noun where adjective needed

l. 84: why mixed tenses?

l. 91: Is integration from -T to T appropriate? If T is the interference time, it appears inappropriate, since -T to T is _twice_ the interference time. (Perhaps this is clarified in one of the references, but I don't have time to read them all, and neither will most readers. Or is it simply a mistake on the part of the authors?)

l. 100: why is it equivalent?

l. 105: "Introduce" - verb where noun needed

Subsequent sentences: why mixed tenses?

l. 108: which acceleration is this?

l. 117: why is Eq. 10 the appropriate form?

l. 129: add citation to Titan accelerometer manufacturer information and documentation (webpage or wherever else the information can be found)

l. 136: define GNSS

Sec. 3.2: "Systematical" is not the preferred word; "systematic" is. This section should be further clarified by adding more details, so that the reader can understand what was done. For example, define "solid tide." How was it calculated? What is the wave vector inversion method? Despite the claim in the text, the offset mentioned here does not seem to be given in Table 1. Explain in the text how the values in Table 1 were obtained.

l. 154: 110 mGal does not appear in Table 1

l. 169: "mooring state" - term not defined. Readers can figure it out from the context, but defining it here would reduce possible confusion.

l. 170: "compensated" - how?

l. 173: "all data have" - but each point in Fig. 4 has a different-sized error bar - this seems to contradict the statement here => need clarification

l. 181: "effect length" - what does this mean? Probably the wrong word was used, and "effective" is what was meant. But why is it "effective" rather than "actual"? => need clarification.

l. 184: "A classical shipborne strapdown gravimeter" - in addition to the Titan accelerometer? Or is the Titan accelerometer what's meant here?

l. 185: "acceleration" - measured by which instrument?

l. 208: the values given here don't correspond to the centers of the dips in Fig. 6a - why not?

l. 211: are the given values before or after improvement via the recovery vector?

Fig. 6a: x-axis incorrectly labeled: "Recover"->"Recovery"; delete "'s"; "component"-->"components"

l. 216: what's meant by "subtracted by 1"?

l. 220: Why are there "anomalies"? What causes them? How are they measured? This is clarified later in the section, but should be clarified at the point of first mention.

Fig. 7: Why do the results after filtering look like a square wave rather than approximately constant? Why is the absolute measurement after filtering approximately g/2 rather than g?

l. 248: "dot"->"dotted"

l. 250: define "inner coincidence"

ll. 257 and 269: "insert"->"inset"; inset x-axis labels missing; if the inset x-axis values are the same as those of the larger figures, the anomalies don't track the sea height. Why is this?

l. 258: procedure described seems circular, thus not valid; "Therefore" - non sequitur; repeated mixing of past and present tenses here and in the following potentially confusing to readers. The steps of this analysis procedure are not clearly described. In going from Fig. 8a to Fig. 8b, the anomaly size increases. Why is this? Why is it an improvement?

l. 262: "Again" - non sequitur; in any case, the following word ("If") should be lowercase.

l. 264: explain why comparing the result using the "optimized" vs nominal recovery vector is the appropriate figure of merit for measurement improvement.

l. 284: "optimizing"->"optimization"

Appendix A is hard to follow - a bit more justification and explanation of the procedures used would help.

Comments on the Quality of English Language

There are many errors of English usage in specialized terminology and in the use or absence of articles ("a," "an," "the"). These should be corrected by someone knowledgeable in English usage. Although I mentioned some of them above, I won't take the time to point them all out here.

Reviewer 2 Report

Comments and Suggestions for Authors

In their manuscript "High precision atom interferometer-based dynamic gravimeter measurement by eliminating the cross-coupling effect” the authors refer to the very well-known problem of operating gravimeters in dynamical environments. The article is in its overall shape professionally written and presents the results in an appropriate and understandable manner. They introduce their method to optimize hybrid inertial sensing between and atom interferometer and a classical accelerometer by eliminating cross-coupling due to misalignment between the sensor axes. The theory behind their method is clearly described. The authors underline the usefulness of their approach by the presentation of measurements, where they indeed could show, that they can increase the combined sensor performance by their method. This work definitely deserve publication and is suitable to be published in the current form.

However, I have small comments for the authors to be considered in the final version:

1.      In lines 125-137 the authors describe their experimental setup. While this is very brief, as they cite a more detailed description of the atomic apparatus, the stabilized platform could be described in more details, since this is a central part of the presented work.

2.      The authors operate their interferometer at T=15 ms, which seems like a decent choice. I wonder, if the authors have some insight (also some data) why they made this choice and how it would influence the contrast, which can be seen in Figure 6, but is not discussed further.

3.      In line 147 the authors refer to a modulation scheme used to generate their Raman laser line. Though this scheme has been used before in other experiments, it might be worth to add a citation.

4.      I believe, this is a problem with the pdf or the draft version, but I anyhow want to address it: the overall display quality of the pictures only allows for borderline visibility, especially thin lines, as the green points in figure 8, are barely visible.

Reviewer 3 Report

Comments and Suggestions for Authors

This article proposes a new method to mitigate the cross-coupling effect that arises in dynamic gravimetric measurements using an AI-based gravimeter coupled to a classical accelerometer. This theoretical method of post-processing nature relies on the introduction and optimization of a recovery vector. It is first presented theoretically, and then experimentally implemented in a maritime gravity survey.

The article is well-written and clearly structured. I believe that the idea is new, original, and has a real potential to enhance the performance of AI-based dynamic gravimetry. Its experimental validation, in a real-life application environment, demonstrates a factor of ~2 improvement in noise reduction, which seems particularly promising. Therefore, I recommend publication of this work in Sensors, provided the below comments are addressed and clarified:

a) From the theoretical section 2., where the accelerations are sometimes denoted as vectors and sometimes scalars, it is not clear which component(s) of acceleration are measured by both the classical accelerometer and the AI-gravimeter. Is it a single-axis or a multi-axis measurement? Clarifying this, and making sure that the entire section 2 is consistent about what each quantity means and how things are denoted, would be needed.

b) On Fig. 4, is there a reason why the error bars after surveying are larger than before surveying?

c)  How was the optimal recovery vector [0.0060;-0.0034;0.9860] obtained? The text suggests that these coordinates correspond to the bottom of the valleys on the calibration curves, but this somehow does not seem compatible with what is displayed on Fig. 6a.

d)   The paper does not mention at all possible non-inertial effects arising in the maritime gravity experiment (arising from ship accelerations, tilts and rotations as induced for instance by waves), and which could affect the dynamics of the atomic cloud – hence inducing systematic effects on the gravity measurements. What is the order of magnitude of these kinematic effects in the experiment? Is there a good reason why the induced systematic effects are negligible here, or have they been mitigated by specific hardware or post-processing? Giving more details about the AI interrogation sequence and gravimeter specifications/parameters would be useful to understand this.

Comments on the Quality of English Language

English language is fine and everything is perfectly understandable.
